# Cancer Pain Assessment and Classification

**DOI:** 10.3390/cancers11040510

**Published:** 2019-04-10

**Authors:** Augusto Caraceni, Morena Shkodra

**Affiliations:** Palliative Care, Pain Therapy and Rehabilitation Department, Fondazione IRCCS-Istituto Nazionale dei Tumori (INT), 20133 Milan, Italy; augusto.caraceni@istitutotumori.mi.it

**Keywords:** cancer pain, pain assessment, pain classification, breakthrough pain, neuropathic pain, pain syndromes

## Abstract

More than half of patients affected by cancer experience pain of moderate-to-severe intensity, often in multiple sites, and of different etiologies and underlying mechanisms. The heterogeneity of pain mechanisms is expressed with the fluctuating nature of cancer pain intensity and clinical characteristics. Traditional ways of classifying pain in the cancer population include distinguishing pain etiology, clinical characteristics related to pain and the patient, pathophysiology, and the use of already validated classification systems. Concepts like breakthrough, nociceptive, neuropathic, and mixed pain are very important in the assessment of pain in this population of patients. When dealing with patients affected by cancer pain it is also very important to be familiar to the characteristics of specific pain syndromes that are usually encountered. In this article we review methods presently applied for classifying cancer pain highlighting the importance of an accurate clinical evaluation in providing adequate analgesia to patients.

## 1. Introduction

In oncological population, pain is one the most invalidating symptoms, affecting approximately 66% of cancer patients [1]. The guidelines for the management of cancer pain were developed by the World Health Organization (WHO) in 1986 [2], but there is substantial evidence that the management of cancer pain is still often suboptimal [3,4,5]. Since cancer pain is not a homogenous entity, correct pain assessment is essential for obtaining satisfactory management. Cancer pain is a general term for a large range of different pain conditions, characterized by different etiology, characteristics, and pathological mechanisms. The importance of adequate pain assessment and complexity of cancer pain has been emphasized for a very long time [6,7]. Considering the importance of pain classification in providing an individual assessment and tailored treatment strategy, over the years there have been a few works that have attempted to find a comprehensive approach to classify cancer pain [8,9,10]. However, no standardized accepted classification system exists yet and different cancer pain classification schemes are used in research and clinical setting.

## 2. Etiology of Pain in Cancer Patients

Not every type of pain in a patient with cancer is related to the tumor and, as a result, not every type of pain perceived by oncological patients can be considered and defined automatically as cancer pain. A prospective study carried on a large sample of oncological patients has shown that ~17% of pain perceived in this group of patients is caused by antineoplastic treatment and approximately 10% due to other etiologies, unrelated to cancer [11]. Therefore, in oncological patients presenting with pain, itis very important to specify if the pain perceived is caused by the tumor, related to treatments or to other comorbidities, in order to be able to provide the necessary treatment.

## 3. Clinical Presentation and Assessment of Pain

Assessment is essential in characterizing pain and identifying the underlying mechanisms, providing a guided decision-making process regarding medical therapy. A comprehensive pain history and clinical examination are both very important for pain assessment. Pain characteristics such as intensity, radiation, duration, temporal variation, qualities, provocative, and alleviating factors, are all crucial for an effective treatment. The use of mnemonics like SOCRATES is useful in clinical setting, providing a systematic approach in assessing pain characteristics (Site, Onset, Character, Radiation, Associated factors, Timing, Exacerbating/relieving factors, and Severity) [12].

Pain characteristics and patients’ characteristics relevant for cancer pain are commonly used in clinical practice to classify or categorize pain in specific domains [13].

### 3.1. Pain Intensity

Intensity is one of the most relevant characteristics of pain, regarded also as the gold standard for pain assessment, which often guides the evaluation and choice of treatment options [14]. Different methods are used to measure intensity, with Numerical Rating Scales (NRS) being one of the most frequent ones. Defining cut points for different levels of pain intensity is important for assessing response to treatment and changes in patient’s status. Several attempts have been made to classify patients according to their pain intensity, for clinical and research purposes. One of the classifications that is used for both these purposes identifies three categories of pain according to the levels of pain severity: mild (NRS 1–4), moderate (5–6), and severe (7–10) [15,16,17]. However, pain intensity needs to be part of a comprehensive assessment and it should be always considered within the individual patient characteristics including age, cognitive function, and psychological aspects.

### 3.2. Pain Site

According to its anatomic location cancer can affect any body tissue, including viscera, bone, soft, and nervous tissue. It is not uncommon for oncological patients, especially when pain is related to metastatic cancer, to have more than one site of pain and this important information is usually recorded using body maps that are included in many assessment tools [18,19,20]. Information should be gathered regarding all pain sites.

### 3.3. Pain Syndromes

Considering the clinical characteristics of pain in cancer patients, based on the recognition of a repeated cluster of signs and symptoms and the relationship of pain with the cancer, it is possible to define some clinical entities which consolidate into specific pain syndromes. The identification of the syndrome can help identify the etiology, prognosis and guide therapeutic interventions [21,22]. Usually the identification of a pain syndrome is based largely on the clinical experience of the clinician, however, over time a few attempts of describing a series of the most common pain syndromes in patients with cancer pain have been made [21,23,24,25]. In Table 1, we report a syndromic classification of cancer pain with anatomical details that was used in an international survey published by the IASP Task Force in cancer pain, which identified some syndromes as most prevalent and suggested that the diagnosis of the primary tumor and pain characteristics were more often related to specific pain syndromes [21].

Pain is a relevant symptom also in patients affected by hematological malignancies. A study of 464 patients affected by hematological cancer revealed 284 different pain syndromes in this group, with 56% diagnosed as deep somatic, 15% as superficial somatic, 14% as visceral, 7% as neuropathic, and 8% mixed. In Table 2 are reported the most common pain syndromes in patients with hematological malignancies [26,27].

### 3.4. Timing and Temporal Variation

Traditionally pain has been classified as acute or chronic, with chronic pain considered as persistent or recurrent pain lasting for a period longer than three months [10]. However, when talking about cancer pain, considering that with the progression of the disease also the damage caused to tissue can progress, it is difficult to make a differentiation between acute and chronic [22].

Regarding temporal variation cancer pain can be either continuous, also known as background pain, or intermittent. In the cancer population, a transitory increment of pain intensity in an otherwise stable chronic pain managed with opioid drugs is common and has been defined as “breakthrough” pain (BTP) [28]. It is estimated that more than one in two patients with cancer pain will also experience BTP, with some variability according to the clinical setting, population, and assessment methods. [29]. BTP is a very heterogeneous entity, and many controversies exist about its definition, epidemiology, and treatment [30,31,32]. In different studies different definitions of BTP are present, including “pain flares”, “incident pain”, “episodic pain”, or “transitory exacerbations” [33,34,35,36]. In a European Association for Palliative Care Research Network Expert Delphi Survey held to reach consensus on definitions, terminology, and classification of transient cancer pain exacerbations, it was concluded that the term ‘transient cancer pain exacerbation’ includes more than just BTP and that pain flares can occur without background pain or with uncontrolled background pain and this phenomena could be named “episodic pain” [37]. Despite the variability observed, there are some key points which are relevant to BTP and are present in all its definitions: the concept of an increased intensity of the background pain with episodic or transient nature of the event, which is often related to a precipitating reversible factor like movement and leads to a need of therapy changes in order to obtain an adequate control of the basal pain. Considering that the presence of this entity has been related to higher baseline pain intensity, identifying BTP in cancer pain patients is relevant for an appropriate pain management scheme [25,38]. It has been shown that BTP can interfere with patients’ normal activities affecting their quality of life [39], and despite its clinical relevance, it is still often inadequately treated. All cancer patients with high pain intensity should be carefully evaluated for the presence of BTP in order to provide optimal pain management and improve patients’ quality of life. Guidelines recommend treating BTP using rapid- or short-acting opioids with rescue doses to help overcome end-of-dose failure [40,41].

## 4. Cancer Pain Mechanisms and Pathophysiology

A recent systematic review revealed that approximately one third of patients do not receive analgesia proportional to their pain intensity [4]. A mechanism-based approach in pain therapy is essential for providing a more individualized and effective analgesia but in order to apply such approach it is important to recognize and classify the underlying pathophysiological processes.

The mechanisms which lead to the development of cancer pain are complicated, and they include a series of changes in cellular, tissue, and systemic levels that occur during tumor proliferation and progression.

In terms of pathophysiological criteria, cancer pain can be classified as either nociceptive, or neuropathic. Nociceptive pain is pain which results from a nociceptor stimulation due to actual or threatened damage of non-neural tissues and can be further classified into somatic and visceral, depending on level of the structures affected [42]. Every pain caused by a lesion or damage of the somatosensory nervous system is considered to be neuropathic [43]. Moreover, cancer pain can often be of mixed pathophysiology, including both a nociceptive and neuropathic component. For example, a nociceptive pain condition can, over time, cause secondary lesions in the somatosensory nervous system leading to the pain in this case being also partly of neuropathic nature.

### Neuropathic Pain

Neuropathic pain (NP) is present in about 19% of patients with cancer pain; 39% if patients with mixed pain are also included [44]. The clinical characteristics of NP are different from those encountered in patients with nociceptive pain and are characterized by the presence of sensory alterations in terms of both hypersensitivity (positive) and hyposensitivity (negative) symptoms and signs. However, determining the presence of NP is not always simple since there is no specific diagnostic tool and no standardized approach to diagnose this type of pain. Based on the combination of pain descriptors such as symptoms, including burning, electric shocks, shooting, pricking, tingling, or pins and needles, and signs like pain evoked by light touching or decreased sensitivity to light touch or pricking, several questionnaires for the screening of NP have been developed. The main aim of these questionnaires, which are mainly based on patient-reported outcomes (PROMs), like LANSS [45], DN4 [46], and painDETECT [20], is to identify patients who may have neuropathic pain and need further assessment, but they cannot be used alone to identify neuropathic pain [43,47]. A recent systematic review on the evaluation of the quality and performance of neuropathic pain assessment tools in identifying neuropathic pain in patients with cancer identified concordance between the clinician diagnosis and screening tool outcomes for LANSS, DN4, and painDETECT [48]. A new study has used a set of patient-reported descriptors associated with nociceptive or neuropathic pain in order to explore sensory symptom profiles in patients with NP, aiming at the development of a new questionnaire, painPREDICT. This questionnaire needs to be further tested, in particular, in cancer patients. Results from the first interim data evidentiated three different characteristic sensory symptom profiles in patients with NP: “irritable nociceptors”, “deafferentation pain”, and “pain attacks with nociceptive component” [49].

In 2008 NeuPSIG proposed a grading system to guide the decision-making process regarding the presence of neuropathic pain [50]. Four criteria were proposed:history of relevant neurological lesion or disease, including pain descriptors, suggestive of pain being related to a neurological lesion and not other causes;pain distribution neuroanatomically plausible;pain associated with the presence of sensory signs in the same neuroanatomically plausible;diagnostic tests confirming a lesion or disease of the somatosensory systems, explaining the pain perceived by the patient.

Three levels of certainty—possible, probable, and definite neuropathic pain—were possible, as described in Figure 1 [44,51].

Still, this grading system has a few limitations: it is obviously based on clinical judgment, and therefore it relies mostly on the experience of the physician and resources available for the assessment and, most importantly, it was not developed to address specifically NP due to cancer [43,51]. A new algorithm, the EAPC/IASP, has been proposed for diagnosing NP in cancer patients [51].

Quantitative sensory testing (QST) is a psychophysical method used to quantify somatosensory function in response to controlled stimuli and it can be useful in providing information about the functional status of somatosensory system but is not recommended as a standalone test for the diagnosis of neuropathic pain [52].

Identifying NP is very important, mostly because it is associated with higher intensities of pain and requires treatment with adjuvant drugs such as glucocorticoids, antidepressants and anticonvulsants, and opioids alone are not sufficient [53].

## 5. Patient Characteristics and Disease Factors

Pain characteristics are all relevant when attempting a systematic classification of pain, however, there are a few other variables, related to the patients’ characteristics and disease that can influence both pain characteristics and response to treatment. Symptoms expression can be influenced by many factor and different studies have demonstrated that domains like psychological distress, sleep disturbances, cognitive function, addictive behavior, age, and primary tumor diagnosis and progression are all related to the complexity of pain condition and can predict the response obtained in different patients [54,55].

## 6. Cancer Pain Classification Systems

Due to the complexity of cancer pain and pain syndromes classifying pain is essential, also because, in particular cases there is a need to introduce different management strategies in order to achieve adequate pain control. Over time, many efforts have been put into bringing together a unique standardized classification system for cancer pain that can be used in both clinical practice and research worldwide. A few classification systems have been developed in order to classify and stratify patients by grouping them according to major common characteristics. However, to date, there is no universally accepted pain classification measure that can accurately predict the prognosis of pain in cancer patients [54,55,56].

### 6.1. International Association for the Study of Pain (IASP) Taxonomy

In 1983, a list of taxonomy on pain was published and based on this taxonomy a classification system was proposed by the IASP [57]. The IASP Classification of Chronic Pain was not aimed at giving a prognosis of pain management but it was developed as a descriptive coding system, for both cancer and noncancer pain. It was based on five axes that are considered relevant for the classification of chronic pain:anatomical region or pain site,system responsible for the pain perceived, temporal characteristics and pattern of pain occurrence,pain intensity and time since onset,etiology.

The aim of the IASP taxonomy was to give a code number to each pain syndrome in order to provide a common language for describing pain, however, despite the updates made in 1994, this approach has not been used widely in clinical practice The use of the IASP Taxonomy has been criticized mainly because it is not aimed at establishing a prognosis and also lacks of assessment of some components that are considered important in cancer pain prognosis. Updates were made to selected sections in 2011 and 2012, mainly regarding pain definitions and terminology [42].

### 6.2. ICD-11

ICD is used for coding different diagnosis in the healthcare system of many countries worldwide. A new IASP Task Force was held in order to provide a new system of classification for chronic pain, resulting in the development of a new classification for the 11th revision of the ICD. The goal was to create a classification system applicable in both primary care and clinical settings for specialized pain management. In this new ICD category for “Chronic Pain” 7 groups were identified, including chronic cancer pain, which is subdivided based on location into visceral, bony (or musculoskeletal), and somatosensory (neuropathic), and is described as either continuous (background pain) or intermittent (episodic pain) [10]. Optional specifiers for recording the time course and severity of the pain as well as the presence of psychosocial factors are included with the recommendations regarding the measurement of cancer-related pain. The cancer-related chronic pain codes are intended to be given as diagnoses of the underlying oncological conditions.

### 6.3. Edmonton Classification System for Cancer Pain (ECS-CP)

An international system for pain classification was developed by Bruera et al. in 1989 and the name of the instrument was Edmonton Staging System (ESS) [58]. It was initially developed as a prognostic indicator for cancer pain management, containing seven domains considered important in achieving adequate pain control: mechanism of pain, incidental pain, daily opioid dose on admission, cognitive function, psychological distress, tolerance, and past history of alcohol or drug addiction. Depending on the combination of these domains, patients were classified as having a good, intermediate or poor prognosis for obtaining adequate pain control. Considering the difficulties in interpreting some of the definitions and aspects of the ESS, a new revised version (rEES) was developed in 2005 by Fainsinger et al. [59]. The rEES contains only five domains: mechanism of pain, incidental pain, psychological distress, addictive behavior, and cognitive function and new definitions for some of the terms. Considering that the aim of this instrument was cancer pain classification the name was changed to Edmonton Classification System for Cancer Pain (ECS-CP) and definitions for the domains like incidental pain, psychological distress, addictive behavior, and cognitive function were modified [60]. The complete ECS-CP is described in Figure 2.

### 6.4. The Cancer Pain Prognostic Scale

The Cancer Pain Prognostic Scale (CPPS) was developed to predict the likelihood of pain relief for cancer patients with moderate to severe pain [61]. It is a predictive formula that includes the worst pain severity, emotional well-being, daily opioid dose, and pain characteristics. The CPPS results can be summarized into a sum from 0 to 17, with higher scores indicating higher probability of pain relief.

## 7. Clinical Relevance

### 7.1. Pain Assessment Clinical Relevance

Pain is a subjective perception and is influenced by both psychosocial- and pathology-related factors. Therefore, the assessment of pain is mainly based on the patients and they should be actively involved in the evaluation process [62]. Considering that cancer pain is often unpredictable and highly variable, an appropriate assessment is essential and should include all aspects of pain [63,64]. In this context, understanding the pathways and mechanisms of cancer pain and being familiar with the different pain syndromes that can be encountered in different tumor types is very important. Pain can be a presenting symptom in cancer patients and might lead to the diagnosis of the disease [65,66,67]. Very often pain can also be a sensitive sign of cancer progression guiding further imaging and testing and helping in cancer staging. For example, a recent study showed that pain referred to the perianal region and painful defecation and weight loss could have a predictive value for locally advanced disease in patients with anal cancer [68]. Hence, when new characteristics or exacerbation of pain are identified, additional attention, and further investigation can be necessary.

### 7.2. Classification Systems Clinical Relevance

Classification systems are very important in order to create a standardized language for pain assessment and clinical work-up but it is unknown how diffused these systems actually are in everyday practice. The IASP Taxonomy was not developed for prognostic purposes and it is composed by an extensive number of pain conditions making its application in everyday practice complicated. The Cancer Pain Prognostic Scale (CPPS) was developed to predict the likelihood of pain relief, however, except for the original study there has been no other report on application and validation of this system. The ESC-CP remains the most widely studied from the classification systems, with prognostic value and assessment of different domains. It has undergone different modifications during the years and is the only classification system that has been validated with findings suggesting that it can predict pain complexity in different practice settings [54]. Despite this, the ESC-CP routine use is still limited. The new ICD-11 with specific codes regarding chronic cancer-related pain is also aimed at providing a standardized assessment regarding cancer pain. However, a wide and correct application in daily practice is strongly related to an appropriate training and providing of adequate information regarding cancer pain assessment and evaluation. There has been a pilot field testing of the classification in which the inclusion of chronic cancer-related pain codes was shown to be strongly welcomed by medical staff [10].

## 8. Conclusions

It is unknown how pain classification systems are actually implemented in clinical practice and in the clinical records of specialized oncology, pain, and palliative care clinics and units. The ECS-CP is regarded as one of the best tools currently available for the classification of cancer pain; however, as mentioned previously, it has not been widely used in clinical practice. This could be mainly due to the lack of a better standardization regarding the assessment of the different domains, part of this classifying system. A simplified version could be less time consuming and could ease the process of completion in everyday practice, but it could also limit the adequate assessment of pain different dimensions. On the other hand, there could be additional domains that could be included in a classification system in order to provide more adequate and reliable information regarding pain as a complex and multidimensional phenomenon. The choice of an appropriate outcome measure is also very important when assessing pain and the prognostic accuracy of a classification system. All these issues have been raised and discussed in an expert conference held in Milan regarding cancer pain assessment and classification which highlighted the core domains such as pain intensity, pain mechanism, breakthrough pain, and psychological distress, and identified some candidate domains and symptoms as relevant in the process of cancer pain classification. These included pain location, genetic variability, and other symptoms such as sleep, depression, and anxiety [69]. New prospective studies will be necessary to provide further information on the association between specific pain outcome measures and different variables, but in order to assess adequately these variables standardized measures and definitions should be provided. A standardized reporting, based on patients’ pain characteristics and underlying mechanisms and a common language used worldwide capable of providing prognostic information for research and clinical purposes would be ideal, leading to better outcomes of both anticancer and antalgic therapy.

## Figures and Tables

**Figure 1 cancers-11-00510-f001:**
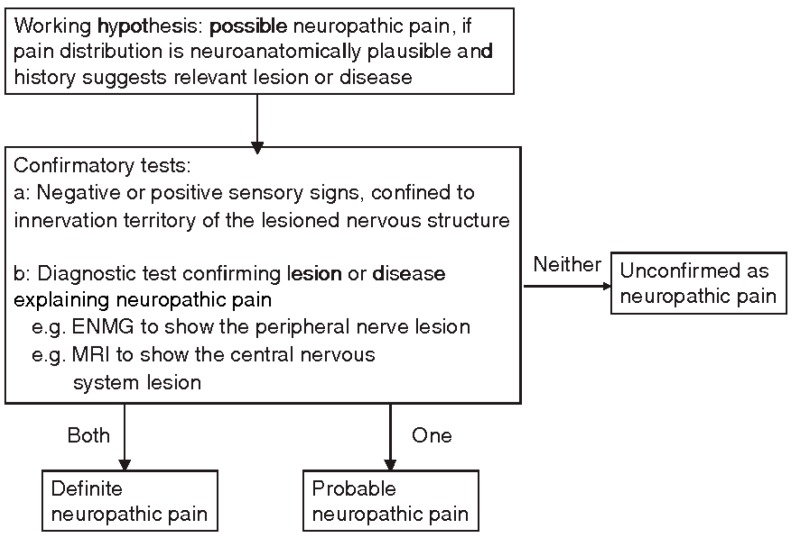
NeuPSIG grading system [50].

**Figure 2 cancers-11-00510-f002:**
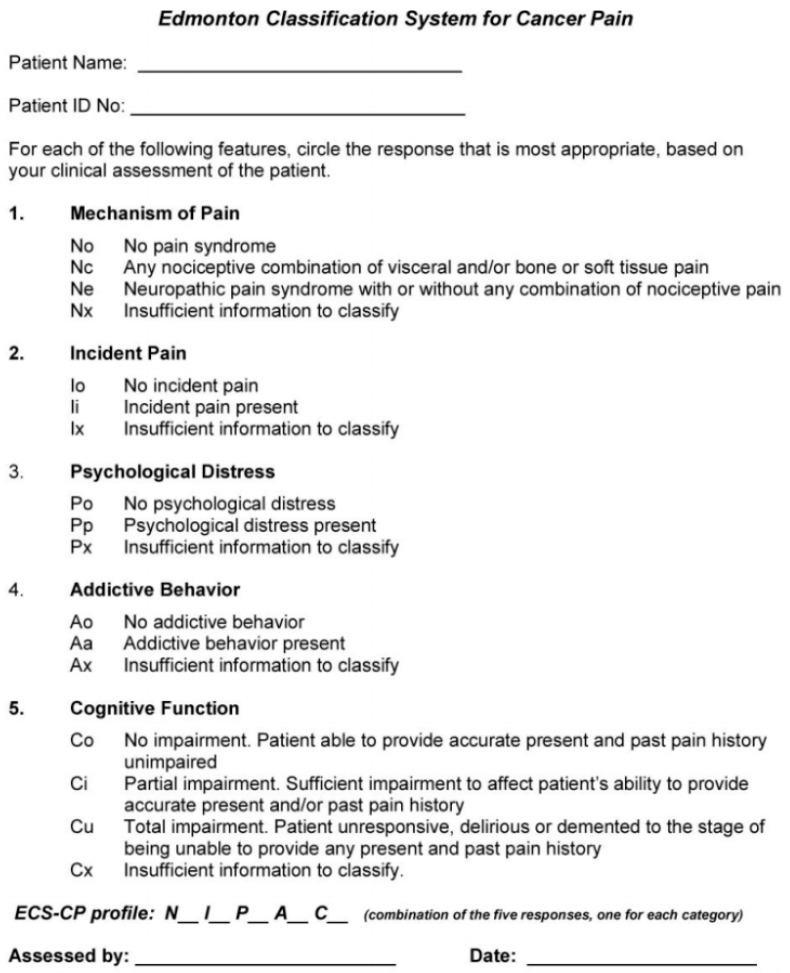
Edmonton Classification System for cancer pain.

**Table 1 cancers-11-00510-t001:** Syndromic classification of pain caused directly by the solid tumor.

Neoplastic damage to bone and joints	1. Base of the skull syndrome. Headache due to calvarial, maxillary, or mandibular lesion
	2. Vertebral syndromes, including sacrum
	3. Pelvic, long bones, direct infiltration of a joint
	4. Generalized bone pain: due to multiple bone metastasis due to bone marrow infiltration/expansion
	5. Chest wall pain from rib lesion
	6. Pathologic fracture of: long bone vertebrae pelvis rib other
Neoplastic damage to viscera	7. Esophageal mediastinal pain.
	8. Shoulder pain from diaphragmatic infiltration - pain from distention of hepatic capsule - obstruction of biliary tract - left upper quadrant pain from splenomegaly
	9. Epigastric pain from pancreas or other upper abdominal neoplasm “Midline rostral retroperitoneal syndrome”
	10. Diffuse abdominal pain from abdominal or peritoneal disease: with obstruction without obstruction
	11. Suprapubic pain from infiltration of bladder. Perineal pain from infiltration of rectum or perirectal tissue (including vagina)
	12. Obstruction of ureter
Neoplastic damage to soft tissue and miscellaneous	13. Damage to oral mucous membranes. Infiltration of skin and subcutaneous tissue
	14. Infiltration of muscle and fascia of in the chest or abdominal wall. Infiltration of muscle and fascia in the limbs
	15. Infiltration of muscle and fascia in the head and neck
	16. Retroperitoneal tissue infiltration excluding rostral retroperitoneal syndrome
	17. Pleural infiltration
Lesions of Nervous Tissue	18. Peripheral nerve syndromes due to paraspinal mass due to chest wall mass due to retroperitoneal mass other than paraspinal due to other soft tissue or bony tumor peripheral polyneuropathy
	19. Radiculopathy or cauda equina syndrome due to vertebral lesion due to leptomeningeal metastases due to other intraspinal neoplasm
	20. Plexopathy cervical plexopathy brachial plexopathy lumbosacral plexopathy sacral plexopathy
	21. Cranial neuropathy due to base of the skull tumor due to leptomeningeal metastases due to other soft tissue or bony cranial tumor
	22. Pain due to central nervous system lesion due to myelopathy intracerebral lesion
	23. Headache due to intracranial hypertension
	24. Neck, back pain or headache due to leptomeningeal disease

**Table 2 cancers-11-00510-t002:** Most common pain syndromes in patients with hematological malignancies.

Pain Type		Pain Origin and Syndromes
Nociceptive	Deep somatic	Bone marrow expansion and osteolysis. Spleen and liver capsulae distension by tumor infiltration and organ enlargement; intracranial hypertension (meningeal and/or brain tumor involvement)
	Superficial somatic	Mucositis, cutaneous lesions
	Visceral	Infiltration and/or compression of viscera cava by abdominal nodes, spleen, and liver enlargement
Neuropathic	Peripheral neuropathic	Neuropathies due to para-proteins. Amyloidosis. Plexopathy by tumor invasion and/or node enlargement compression (lymphomas)
	Central Neuropathic	CNS damage and/or tumor involvement
Mixed	Neuropathic + somatic	Meningosis, peripheral nerve damage, and/or tumor involvement

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
