# Peer review of "Cancer Pain Assessment and Classification"

_cancers, 2019, doi:10.3390/cancers11040510_

Reviewer 1 Report

The authors wrote a pleasant, relevant and well organized article.

However, one paragraph should be added about the causes that could explain the reluctance to use most of these classifications. Authors could give some proposals to improve the use of these kind of classifications.

Second point, authors could increase the impact of their review if they described the advantages and weaknesses of each classification and specify if these classifications can be use as prognostic or predictive factors.

Otherwise, this review is very informative and deserves to be published

Author Response

Thank you for the appreciation of our work and the kind words.                   

Point 1: One paragraph should be added about the causes that could explain the reluctance to use most of these classifications. Authors could give some proposals to improve the use of these kind of classifications.

Response 1:

Thank you for your suggestion. In the updated version of the manuscript we have extended the section “8. Conclusions” page 10, by adding as recommended some explanations on why classification systems are not widely used in clinical practice and we have added some suggestions on how this could be improved in the future, as follows:

It is unknown how pain classification systems are actually implemented in clinical practice and in the clinical records of specialized oncology, pain and palliative care clinics and units. Undertreatment in patients affected by cancer pain is still frequent, and often associated with inadequate assessment and lack of appropriate classification. The ECS-CP is regarded as one of the best tools currently available for the classification of cancer pain, however, as mentioned previously it has not been widely used in clinical practice. This could be mainly due to the lack of a better standardization regarding the assessment of the different domains, part of this classifying system. A simplified version could be less time consuming and could ease the process of completion in everyday practice, but it could also limit the adequate assessment of pain different dimensions. On the other hand, there could be additional domains that could be included in a classification system in order to provide more adequate and reliable information regarding pain as a complex and multidimensional phenomenon. The choice of an appropriate outcome measure is also very important when assessing pain and the prognostic accuracy of a classification system. All these issues have been raised and discussed in an expert conference held in Milan regarding cancer pain assessment and classification which highlighted the core domains such as pain intensity, pain mechanism, breakthrough pain and psychological distress and identified some candidate domains and symptoms as relevant in the process of cancer pain classification. These included pain location, genetic variability and other symptoms such as sleep, depression and anxiety [70]. New prospective studies will be necessary to provide further information on the association between specific pain outcome measures and different variables, but in order to assess adequately these variables standardized measures and definitions should be provided. A standardized reporting, based on patients’ pain characteristics and underlying mechanisms and a common language used worldwide capable of providing prognostic information for research and clinical purposes would be ideal, leading to better outcomes of both anticancer and antalgic therapy.

Point 2: Authors could increase the impact of their review if they described the advantages and weaknesses of each classification and specify if these classifications can be use as prognostic or predictive factors.

Response 2: When explaining each of the classification systems we have provided information about the predictive or prognostic values however following your suggestion we have added a new section “7.2. Classification systems clinical relevance” page 10, as follows:

7.2. Classification systems clinical relevance

Classification systems are important in order to create a standardized language for pain assessment and clinical work-up but it is unknown how differed these systems actually are in everyday practice. The IASP Taxonomy was not developed for prognostic purposes and it is composed by an extensive number of pain conditions making its application in everyday practice complicated. The Cancer Pain Prognostic Scale (CPPS) was developed to predict the likelihood of pain relief however except for the original study there has been no other report on application and validation of this system. The ESC-CP remains the most widely studied from the classification systems, with prognostic value and assessment of different domains. It has undergone different modifications during the years and is the only classification system that has been validated with findings suggesting that it can predict pain complexity in different practice settings [55]. Despite this, the ESC-CP routine use is still limited. The new ICD-11 with specific codes regarding chronic cancer-related pain is also aimed at providing a standardized assessment regarding cancer pain. However, a wide and correct application in daily practice is strongly related to an appropriate training and providing of adequate information regarding cancer pain assessment and evaluation. There has been a pilot field testing of the classification in which the inclusion of chronic cancer-related pain codes was shown to be strongly welcomed by medical staff [10].

Reviewer 2 Report

Congratulations on the article, it does a great work summing up the assessment of pain in oncology, and there are not an excess of articles on this topic. However, some aspects should be corrected: 

1.- The bibliography, especially the initial one, is very old and there are more recent articles, it should be updated 

2.- The classification should be similar for solid tumors and for hematological tumors. 

In the solid tumors in the classification, the pain produced by the treatment is excluded and in the hematological tumors is included (example: mucositis)

In the classification of hematological tumors, the classification is based on the pathophysiology and solid tumors are not included. 

In hematological classification: Breakthrough pain is not a physiopathological classification (it can be neuropathic, nociceptive, mixed ...)

Breakthrough pain is to a type of pain in solid tumors

3.- In the neuropathic pain section If there is a recent meta-analysis that evaluates the diagnostic tools in neuropathic pain in cancer (Mulvey 2017) 

Has the PAINpredict been tested in cancer patients? If not, it should be specified

Author Response

Thank you for the appreciation of our work and the kind words.

Point 1: The bibliography, especially the initial one, is very old and there are more recent articles, it should be updated.

Response 1: Thank you for pointing this out. We have updated in the revised version the first part of the bibliography, specifically citation 3,4&5 in the first manuscript, with more recent articles regarding the suboptimal management of cancer pain as suggested.

·          Greco, M.T.; Roberto, A.; Corli, O.; Deandrea, S.; Bandieri, E.; Cavuto, S.; Apolone, G. Quality of Cancer Pain Management: An Update of a Systematic Review of Undertreatment of Patients with Cancer. Journal of clinical oncology 2014, 32, 4149-4154.

·          Kwon, J.H. Overcoming Barriers in Cancer Pain Management. Journal of Clinical Oncology 2014, 32, 1727-1733.

·          Reis-Pina, P.; Lawlor, P.G.; Barbosa, A. Adequacy of Cancer-Related Pain Management and Predictors of Undertreatment at Referral to a Pain Clinic. J. Pain Res. 2017, 10, 2097-2107.

We have also updated reference n. 10 in the first manuscript with the most recent paper regarding ICD-11 classification of cancer pain as follows:

·          Bennett, M.I.; Kaasa, S.; Barke, A.; Korwisi, B.; Rief, W.; Treede, R.D.; IASP Taskforce for the Classification of Chronic Pain. The IASP Classification of Chronic Pain for ICD-11: Chronic Cancer-Related Pain. Pain 2019, 160, 38-44.

Regarding references 6 and 7 in the first manuscript:

·          Foley, K.M. Pain syndromes in patients with cancer. In Cancer Pain.;Springer, 1987, pp. 45-54.

·          Bonica, J.J. The Management of Pain. Am. J. Med. Sci. 1954, 227, 593.

We think that they were necessary in order to emphasize the historical point of view of the idea that we wanted to express.

The importance of adequate pain assessment and complexity of cancer pain has been emphasized for a very long time [6,7]. “

As for reference 11:

·          Grond, S.; Zech, D.; Diefenbach, C.; Radbruch, L.; Lehmann, K.A. Assessment of Cancer Pain: A Prospective Evaluation in 2266 Cancer Patients Referred to a Pain Service. Pain 1996, 64, 107-114.

 The article cited is one of the only prospective studies regarding pain assessment, which has included 2266 consecutive cancer patients therefore we think it should be part of the bibliography.

Point 2: The classification should be similar for solid tumors and for hematological tumors. 

In the solid tumors in the classification, the pain produced by the treatment is excluded and in the hematological tumors is included (example: mucositis)

In the classification of hematological tumors, the classification is based on the pathophysiology and solid tumors are not included. 

In hematological classification: Breakthrough pain is not a physiopathological classification (it can be neuropathic, nociceptive, mixed ...)

Breakthrough pain is to a type of pain in solid tumors

Response 2: Thank you for raising these important points. We agree that the classification should be similar for both solid and hematological tumors. However, there is actually very little information in the literature regarding the syndromic classification of haematological tumors. The table we have described is from the only paper we found providing a standardised description of pain syndromes in patients affected by haematological malignancies. It is true that breakthrough pain is not a physiopathological classification of pain therefore we have modified Table 2 in the revised version of the manuscript by deleting this part.

Regarding the solid tumors syndromic classification we intentionally listed only the syndromes of pain related directly to the solid tumour and not treatement-related. We have modified the heading of the table 1 from “Syndromic classification of Cancer Pain” into “Syndromic classification of pain caused directly by the solid tumour” in the revised manuscript in order for this to be more clear.

Point 3: In the neuropathic pain section If there is a recent meta-analysis that evaluates the diagnostic tools in neuropathic pain in cancer (Mulvey 2017) . Has the PAINpredict been tested in cancer patients? If not, it should be specified.

Response 3: You are right. There is a very good meta-analysis evaluating different tools used for the screening of neuropathic pain component in patients affected by cancer pain. We have added, as suggested the following reference to the revised version of the manuscript and its results:

49.  Mulvey, M.; Boland, E.; Bouhassira, D.; Freynhagen, R.; Hardy, J.; Hjermstad, M.; Mercadante, S.; Pérez, C.; Bennett, M. Neuropathic Pain in Cancer: Systematic Review, Performance of Screening Tools and Analysis of Symptom Profiles. BJA: British Journal of Anaesthesia 2017, 119, 765-774.

Regarding painPREDICT, it is a new questionnaire under development. The article cited reports the first interim data from the development of this questionnaire therefore it has been developed in non-cancer patients and not yet tested yon any population of patients. As requested, in the new revised version we have specified this as follows:

“A new study has used a set of patient-reported descriptors associated with nociceptive or neuropathic pain in order to explore sensory symptom profiles in patients with NP, aiming at the development of a new questionnaire, painPREDICT. This questionnaire needs to be further tested in particular in cancer patients. Results from the first interim data evidentiated three different characteristic sensory symptom profiles in patients with NP: "irritable nociceptors", "deafferentation pain" and "pain attacks with nociceptive component”

Reviewer 3 Report

Dear Authors,

Thank you very much for giving me the opportunity to review the manuscript. This is a very relevant and important topic. I found the manuscript well written, clear and comprehensive. I recommend it for publication with minor suggestions for improvement:

Section 3.1 Pain Intensity

I think it is important to mention the shortcomings (described in the literature) of the pain intensity scores, especially when managing chronic cancer pain. How a patient's perception of their pain/disease, global suffering, psychological distress etc. may affect the reporting and how overreliance on the numerical pain intensity score could lead to pharmacological overtreatment.

2. Table 1:

-  under "neoplastic damage to bone and joints", point 6: correct the spelling "Pathologic"

- Under "lesions of nervous tissue" #18, I would consider adding peripheral nerve entrapment syndromes.

3. Conclusions:

The authors conclude that the ECS-CP score is one of the best tools available for the classification of cancer pain. However, it has been more than 10 years since the publication of the original research and it has not been widely accepted/implemented in the clinical practice. I think it would be important to mention why this has not happened and perhaps recommend steps towards wider implementation of the system.  

Author Response

Thank you for your time. We appreciate the comments and the kind words.

Point 1: Section 3.1 Pain Intensity

I think it is important to mention the shortcomings (described in the literature) of the pain intensity scores, especially when managing chronic cancer pain. How a patient's perception of their pain/disease, global suffering, psychological distress etc. may affect the reporting and how overreliance on the numerical pain intensity score could lead to pharmacological overtreatment.

Response 1: Thank you for pointing this out. We agree with your statement. There are indeed shortcomings in using pain intensity scores and they cannot always be reliable but subjective pain assessment is however a necessary and relevant component of the clinical decision process. We have added to the section 3.1 Pain intensity page 2 as suggested a statement to emphasize the idea of global pain:

“However, pain intensity needs to be part of a comprehensive assessment and it should be always considered within the individual patient characteristics including age, cognitive function and psychological aspects.”

Furthermore, it also has been mentioned in sections 5. “Patient characteristics and disease factors” and 7.1. “Pain clinical relevance” that pain is a multidimensional entity influenced by many different aspects including patient and disease characteristics.

Point 2: Table 1:

-  under "neoplastic damage to bone and joints", point 6: correct the spelling "Pathologic"

- Under "lesions of nervous tissue" #18, I would consider adding peripheral nerve entrapment syndromes.

Response 2: Thank you for pointing out the mistake in the spelling. We have corrected it in the revised version. Regarding the second point, in table 1 we have described the syndromic classification of cancer pain reported in the international survey published by the IASP Task Force in cancer pain, which identified these syndromes as the most prevalent. In this very general systemic classification a tumor nerve entrapment would be classified together with other types if nervous tissue lesions.

Point 3: Conclusions:

The authors conclude that the ECS-CP score is one of the best tools available for the classification of cancer pain. However, it has been more than 10 years since the publication of the original research and it has not been widely accepted/implemented in the clinical practice. I think it would be important to mention why this has not happened and perhaps recommend steps towards wider implementation of the system.  

Response 3: Thank you for your comment. Actually, also reviewer 1 raised this point and therefore we decided to modify sections 8. “Conclusions” in the new edited manuscript by adding some of the reasons why we think the ECS-CP score has not been widely implemented in the clinical practice and also recommending some steps that could be useful in reaching this objective as follows:

It is unknown how pain classification systems are actually implemented in clinical practice and in the clinical records of specialized oncology, pain and palliative care clinics and units. Undertreatment in patients affected by cancer pain is still frequent, and often associated with inadequate assessment and lack of appropriate classification. The ECS-CP is regarded as one of the best tools currently available for the classification of cancer pain, however, as mentioned previously it has not been widely used in clinical practice. This could be mainly due to the lack of a better standardization regarding the assessment of the different domains, part of this classifying system. A simplified version could be less time consuming and could ease the process of completion in everyday practice, but it could also limit the adequate assessment of pain different dimensions. On the other hand, there could be additional domains that could be included in a classification system in order to provide more adequate and reliable information regarding pain as a complex and multidimensional phenomenon. The choice of an appropriate outcome measure is also very important when assessing pain and the prognostic accuracy of a classification system. All these issues have been raised and discussed in an expert conference held in Milan regarding cancer pain assessment and classification which highlighted the core domains such as pain intensity, pain mechanism, breakthrough pain and psychological distress and identified some candidate domains and symptoms as relevant in the process of cancer pain classification. These included pain location, genetic variability and other symptoms such as sleep, depression and anxiety [70]. New prospective studies will be necessary to provide further information on the association between specific pain outcome measures and different variables, but in order to assess adequately these variables standardized measures and definitions should be provided. A standardized reporting, based on patients’ pain characteristics and underlying mechanisms and a common language used worldwide capable of providing prognostic information for research and clinical purposes would be ideal, leading to better outcomes of both anticancer and antalgic therapy.

Round  2

Reviewer 1 Report

Authors improved their article. Now definitively acceptable